# VTranM: Vision Transformer Explainability with Vector Transformations Measurement

## Abstract

While Vision Transformers, characterized by their growing complexity, excel in various computer vision tasks, the intricacies of their internal dynamics remain largely unexplored. To embed visual information, Vision Transformers draw representations from image patches as transformed vectors and subsequently integrate them using attention weights. However, current explanation methods only focus on attention weights without considering essential information from the corresponding transformed vectors, failing to accurately illustrate the rationales behind models' predictions. To accommodate contributions of transformed vectors, we propose **VTranM**, a novel explanation method leveraging our introduced vector transformation measurement. Specifically, our measurement faithfully evaluates transformation effects by considering changes in vector length and directional correlation. Furthermore, we use an aggregation framework to incorporate attention and vector transformation information across layers, thus capturing the comprehensive vector contributions over the entire model. Experiments on segmentation and perturbation tests demonstrate the superiority of VTranM compared to state-of-the-art explanation methods.

## 1 Introduction

Transformers have seen surging popularity in various computer vision tasks [45, 9, 15, 44, 22, 28]. The application of these models has resulted in superior performance, paving the way for new breakthroughs in numerous domains. However, the inherent complexity of Vision Transformers often renders them black-box models, making understanding their internal mechanisms a challenge. Such a lack of transparency undermines the trustworthiness of decision-making processes [24, 47, 36, 14, 2]. Therefore, it is required to interpret Transformer networks. Post-hoc explanation is an effective method to provide human-understandable interpretations, which elucidates the rationale behind the model's prediction by evaluating input elements' contributions.

There are two types of post-hoc explanation methods, general traditional explanations [35, 43] and Transformer-specific attention-based explanations [1, 10]. Although traditional explanations excel in interpreting MLPs and CNNs, their efficacy is significantly diminished when applied to Vision Transformers, due to the fundamental differences in model structures. As a response, attention-based explanations develop new paradigms specific to Transformers, where attention weights play a dominant role [1, 11, 10, 33]. Integrating the useful attention information, these methods offer a more promising approach toward Transformer explainability.

Recent advancements in attention-based explanations have led to interesting insights into Vision Transformers. However, the inherent complexity of these models, especially with their core modules: Multi-Head Self-Attention (MHSA) and Feed Forward Network (FFN) [15], still presents distinct challenges in explainability. As elucidated in Section 2, we represent the outputs of both MHSA and FFN as weighted sums of transformer vectors, with each vector scaled by an attention weight. Existing attention-based methods simply consider the scaling weights indicated by attention maps as the corresponding vectors' contributions, overlooking the impacts of vector transformations. As shown in Figure 1, attention weights alone misrepresent the contributions from foreground objects or background regions, while transformation information offers a necessary counterbalance. For instance, even if certain background regions are scaled by high attention weights, their actual contribution can be diminished if they are transformed into smaller or divergent vectors. Conversely,

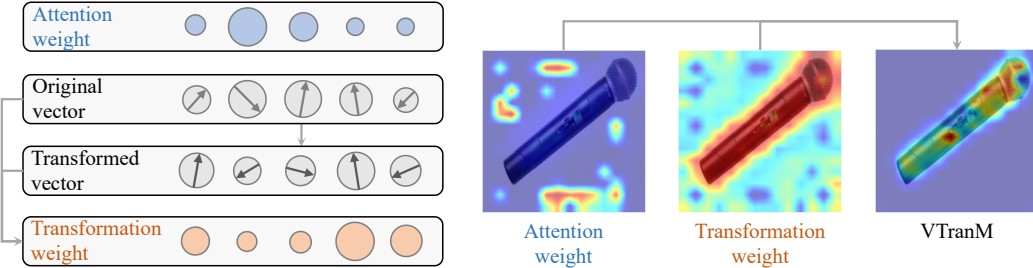

Figure 1: Visualization of attention and transformation weights, and the result of our VTranM that integrates both of them. Circle sizes signify weight magnitudes or vector lengths, and arrows indicate directions. Transformation weights are derived by our proposed measurement, which evaluates the transformation effects by gauging changes in length and direction. Both weights are visualized by heatmaps. Solely using attention weights often fails to localize foreground objects and inaccurately highlights noisy backgrounds. In contrast, leveraging additional information from vector transformation, the VTranM produces object-centric explanations.

a foreground object, despite receiving minimal attention weights, can play a pivotal role due to significant transformation within the model. Given the intertwined dynamics of attention and vector transformations, there is an imperative need for a comprehensive explanation method that cohesively addresses both elements.

In this paper, we propose **VTranM**, a novel explanation method for Vision Transformers using vector transformation measurement. We first reinterpret Vision Transformer layers from a generic perspective, which conceives both MHSA and FFN's outputs as weighted linear combinations of original and transformed vectors. Intuitively, vectors with increased magnitude and aligned orientations will significantly influence the linear combination outcomes. Therefore, to quantify the effects of transformations, our introduced measurement focuses on two fundamental vector attributes: length and direction. Using the proposed measure, we integrate transformation effects, quantified by transformation weights, with the attention information, ensuring a faithful assessment of vector contributions. Moreover, Vision Transformers consist of sequentially stacked layers, each performing vital vector transformations and globally mixing different vectors, a process termed contextualization [45]. Recognizing the accumulative nature of these mechanisms, a single-layer analysis remains insufficient [8, 1]. To holistically evaluate all layers, our solution encompasses an aggregation framework. Employing rigorous initialization and update rules, our framework yields a comprehensive contribution map, which reveals vector contributions over the entire model. Experiments on segmentation and perturbation tests show that our VTranM outperforms state-of-the-art methods.

In summary, our contributions are as follows: **(i)** We explore Vision Transformer explainability and identify a primary issue: the lack of comprehensive consideration for vector transformations and attention weights, which can result in misleading interpretations of the model's prediction. **(ii)** We introduce VTranM, a novel explanation method employing a vector transformation measurement. This measurement regards changes in vector length and directional correlation, which faithfully assesses the impact of transformations on vector contributions. **(iii)** Our approach establishes an aggregation framework that integrates both attention and vector transformation information across multiple layers, thereby capturing the cumulative nature of Vision Transformers. **(iv)** Using the proposed measurement and framework, our VTranM demonstrates superior performance in comparison to existing state-of-the-art methods.

## 2 MOTIVATION

Vision Transformer [15] sequentially stacks $n_L$ layers, each embodying an MHSA or an FFN. We revisit these layers generically, then analyze the problem in Vision Transformer explanations.

### 2.1 REINTERPRETING TRANSFORMER LAYERS

From a general viewpoint, MHSA and FFN both process weighted linear combinations of original and transformed vectors. Starting with a reinterpretation of MHSA, we illustrate this shared principle, then we show how FFN emerges as a particular case within the unified formulation.

In MHSA, every vector of the input sequence attends to all others by projecting the embeddings to a query, key, and value. Formally, let $\mathbf{E} \in \mathbb{R}^{n \times d}$ be the embeddings matrix (a sequence of vectors), where $n$ is the number of vectors, and $d$ is the dimensionality of embedding space. The projections can be expressed as:

$$\mathbf{Q} = \mathbf{E}\mathbf{W}^{\mathbf{Q}}, \quad \mathbf{K} = \mathbf{E}\mathbf{W}^{\mathbf{K}}, \quad \mathbf{V} = \mathbf{E}\mathbf{W}^{\mathbf{V}}, \tag{1}$$

where $\mathbf{W}^{\mathbf{Q}} \in \mathbb{R}^{d \times d_Q}$, $\mathbf{W}^{\mathbf{K}} \in \mathbb{R}^{d \times d_K}$, and $\mathbf{W}^{\mathbf{V}} \in \mathbb{R}^{d \times d_V}$ are parameter matrices. Subsequently, the attention map $\mathbf{A}$ is computed by:

$$\mathbf{A} = \text{Softmax}\left(\frac{\mathbf{Q}\mathbf{K}^T}{\sqrt{d_Q}}\right) \in \mathbb{R}^{n \times n}, \tag{2}$$

Then, contextualization is performed on value $\mathbf{V}$ using attention map $\mathbf{A}$, and another linear transformation further projects the resulting embeddings back to the space $\mathbb{R}^d$:

$$(\mathbf{A}\mathbf{V})\mathbf{W}^{\mathbf{H}}, \tag{3}$$

where $\mathbf{W}^{\mathbf{H}} \in \mathbb{R}^{d_V \times d}$. To obtain the final output, MHSA integrates the results from multiple heads and incorporates them into original embeddings $\mathbf{E}$ from the previous layer by skip-connection [21]. The output of MHSA is given by:

$$\text{MHSA}(\mathbf{E}) = \mathbf{E} + \sum_{h=1}^{n_H} (\mathbf{A}\mathbf{V})\mathbf{W}^{\mathbf{H}}, \tag{4}$$

where $n_H$ is the number of heads. We omit the subscript $h$ for simplicity. Given the associative property of matrix multiplication, we can regard the combination of value projection and multi-head integration as a simple yet equivalent vector transformation featured by $\widetilde{\mathbf{W}} = \mathbf{W}^{\mathbf{V}}\mathbf{W}^{\mathbf{H}}$:

$$(\mathbf{A}\mathbf{V})\mathbf{W}^{\mathbf{H}} = \mathbf{A}(\mathbf{E}\mathbf{W}^{\mathbf{V}})\mathbf{W}^{\mathbf{H}} = \mathbf{A}\mathbf{E}(\mathbf{W}^{\mathbf{V}}\mathbf{W}^{\mathbf{H}}) = \mathbf{A}(\mathbf{E}\widetilde{\mathbf{W}}). \tag{5}$$

We then reformulate the MHSA using the definition of transformed embeddings $\widetilde{\mathbf{E}} = \mathbf{E}\widetilde{\mathbf{W}}$:

$$\text{MHSA}(\mathbf{E}) = \mathbf{E} + \sum_{h=1}^{n_H} \mathbf{A}\widetilde{\mathbf{E}}. \tag{6}$$

From a vector-level view, each vector of the MHSA's output is a weighted sum of original and transformed vectors. With this insight, the FFN embodies a basic form of 'contextualization', where the number of heads $n_H = 1$ and the attention map $\mathbf{A} = \mathbf{I}$ (identity matrix), as contrasted with Equation 6. Utilizing the concept of transformed embeddings $\widetilde{\mathbf{E}}$, the FFN can be expressed as:

$$\text{FFN}(\mathbf{E}) = \mathbf{E} + \mathbf{I}\widetilde{\mathbf{E}}, \tag{7}$$

where $\widetilde{\mathbf{E}}$ consists of vectors that have been transformed within the FFN.

## 2.2 PROBLEM IN EXPLAINING VISION TRANSFORMERS

In our reformulations, as depicted in Equation 6 and Equation 7, Vision Transformer layers are generically expressed as weighted sums of original and transformed vectors, where each vector is multiplied by a scaling weight. Existing attention-based methods typically account for these scaling weights as the corresponding vectors' contributions, which overlooks the influences imparted by the vectors. As illustrated in Figure 1, relying solely on attention weights can misrepresent the contributions from various image patches. This issue necessitates a comprehensive method to faithfully interpret the inner mechanism of Transformer layers.

## 3 METHOD

In this section, we propose VTranM. We first introduce the background of attention-based explanations for Vision Transformers. Then, we develop a vector transformation measurement, which evaluates the effects of transformed vectors. Finally, we design an aggregation framework to accumulate both attention and transformation information across all layers.

### 3.1 ATTENTION-BASED EXPLANATIONS

Attention-based explanation methods [1, 11, 10] measure the contribution of each vector using the attention information, as expressed by:

$$\mathbf{C} = \mathbf{O} + \mathbb{E}_{\mathrm{h}}\left[(\nabla_{\mathbf{A}}p(c))^+ \odot \mathbf{T}\right], \quad \text{with} \quad \mathbf{O} = \mathbf{I}, \quad \mathbf{T} = \mathbf{A}. \tag{8}$$

Here, $\odot$ is the Hadamard product. $\mathbf{C} \in \mathbb{R}^{n \times n}$ denotes the contribution map, where $C_{ij}$ represents the influence of the $j$-th input vector on the $i$-th output vector. Matrices $\mathbf{O} \in \mathbb{R}^{n \times n}$ and $\mathbf{T} \in \mathbb{R}^{n \times n}$ reflect the contributions from the original and transformed vectors, respectively. Previous explanation methods quantify $\mathbf{O}$ and $\mathbf{T}$ simply using scaling weights. Specifically, $\mathbf{I}$ is an identity matrix representing the self-contributions of original vectors, and $\mathbf{A}$ is the attention weights for scaling transformed vectors. Moreover, $\nabla_{\mathbf{A}}p(c) = \frac{\partial p(c)}{\partial \mathbf{A}}$ is the partial derivative of attention map *w.r.t.* the predicted probability for class $c$. $\mathbb{E}_{\mathrm{h}}$ is the mean over multiple heads. Note that it averages across heads using the gradient $\nabla_{\mathbf{A}}p(c)$ to become class-specific, and it removes the negative contributions before averaging to avoid distortion [46, 6, 11, 10]. The previous methods' limitation lies in their assumption of equal influences from the skip connection and the attention weights, neglecting the difference between original and transformed vectors.

### 3.2 VECTOR TRANSFORMATION MEASUREMENT

To account for the contributions from transformed vectors, we introduce a measurement that gauges the influence of vector transformations and subsequently derives transformation weights. Using these weights, we recalibrate the matrices $\mathbf{O}$ and $\mathbf{T}$ to encapsulate both attention and transformation insights. Since MHSA and FFN are depicted in a unified form (see Section 2), we can first derive our measurement based on MHSA, and then adapt it for FFN from a generic perspective.

Drawing from foundational principles of vector representations, we emphasize two core attributes: length and direction [42, 25]. The core reasoning is that vectors with greater lengths and consistent spatial orientations predominantly determine the results of linear combinations. Thus, our measurement will involve two components, corresponding to these two attributes. The first component is a length function $\mathrm{L}(\mathbf{x}):\mathbb{R}^d \to \mathbb{R}^+$, which measures the length of a vector, whether the original or transformed. Mathematically, we instantiate L using $L^2$ norm of the embedding space $\mathbb{R}^d$, *i.e.,* $\mathrm{L}(\mathbf{x}) = \|\mathbf{x}\|_2$. Considering both the vector length measurement L and the attention weights, we reintroduce $\mathbf{O}$ and $\mathbf{T}$ as:

$$\mathbf{O} = \mathbf{I} \cdot \mathrm{diag}(\mathrm{L}(\mathbf{E}_1), \mathrm{L}(\mathbf{E}_2), \dots, \mathrm{L}(\mathbf{E}_n)), \tag{9}$$

$$\mathbf{T} = \mathbf{A} \cdot \mathrm{diag}(\mathrm{L}(\widetilde{\mathbf{E}}_1), \mathrm{L}(\widetilde{\mathbf{E}}_2), \dots, \mathrm{L}(\widetilde{\mathbf{E}}_n)). \tag{10}$$

Note that $C_{ij}$ indicates the contribution of the $j$-th input vector. Thus we apply the function L column-wise. This approach accounts for both the attention weights and the vectors themselves. To evaluate the changes in contributions after transformations, we now regard the original vectors as reference units and analyze the relative effects of transformations. This is done by normalizing $\mathbf{O}$ and $\mathbf{T}$ column-wise *w.r.t.* the lengths of the original vectors. As a result, we obtain:

$$\mathbf{O} = \mathbf{I}, \quad \mathbf{T} = \mathbf{A} \cdot \mathrm{diag}\left(\frac{\mathrm{L}(\widetilde{\mathbf{E}}_1)}{\mathrm{L}(\mathbf{E}_1)}, \frac{\mathrm{L}(\widetilde{\mathbf{E}}_2)}{\mathrm{L}(\mathbf{E}_2)}, \dots, \frac{\mathrm{L}(\widetilde{\mathbf{E}}_n)}{\mathrm{L}(\mathbf{E}_n)}\right). \tag{11}$$

In this formulation, $\mathbf{O}$ uses the identity matrix to represent the contributions from original vectors as basic reference units. Meanwhile, $\mathbf{T}$ discerns the relative influences of transformed vectors compared to the original vectors using the ratio of their lengths. As detailed in Section 3.3, this approach allows us to initialize a contribution map with the lengths of the model's input vectors, and then iteratively update the map across layers. The update employs the ratios of vector effects between consecutive layers, tracing the evolution of transformations within the model. This framework ensures that our analysis remains grounded to the initial input to the model, yet dynamically adapts to every vector transformation and contextualization encountered in the model.

Our second component focuses on directions. Beyond adjusting lengths, vector transformations also influence directions [16]. A transformed vector that stays directionally closer in representation space is expected to have a stronger correlation with its original counterpart. To quantify this directional

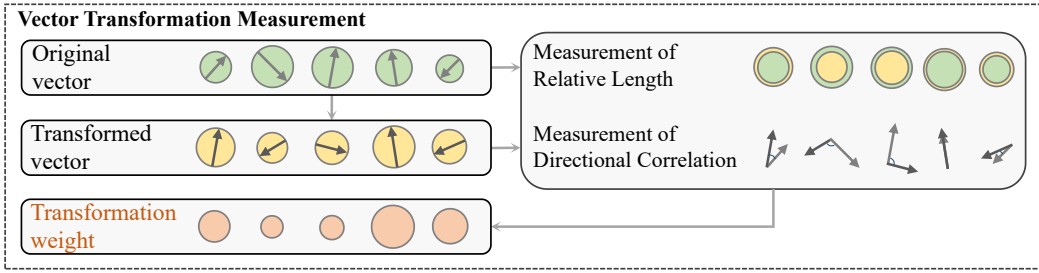

Figure 2: Illustration of our vector transformation measurement. We depict original and transformed vectors with circles and arrows. Circle sizes reflect lengths, and arrows denote directions. The effects of vector transformation are reflected by the changes in length and direction. Our method considers both properties to evaluate these effects, resulting in the corresponding transformation weights.

correlation, we introduce a function $C(\mathbf{x}, \widetilde{\mathbf{x}}):\mathbb{R}^d \times \mathbb{R}^d \rightarrow \mathbb{R}$. Mathematically, we employ Cosine similarity, which measures the angle between a pair of vectors, *i.e.*, $C(\mathbf{x}, \widetilde{\mathbf{x}}) = \cos\langle \mathbf{x}, \widetilde{\mathbf{x}} \rangle$. Next, we will use the function C to complement the length factors in matrix $\mathbf{T}$. Simply multiplying C as a coefficient may introduce negative values to the contribution map, which will distort the signs of contributions through aggregation [40, 35, 6, 11, 10]. Inspired by [27], we propose the Normalized Exponential Cosine Correlation (NECC). This measurement is normalized to emphasize the relative magnitudes of each correlation instead of the polarity, thereby serving as an effective positive weighting factor. For a sequence of original vectors $\mathbf{S_E} = (\mathbf{E}_1, \mathbf{E}_2, \ldots, \mathbf{E}_n)$ and their transformed counterparts $\mathbf{S}_{\widetilde{\mathbf{E}}} = (\widetilde{\mathbf{E}}_1, \widetilde{\mathbf{E}}_2, \ldots, \widetilde{\mathbf{E}}_n)$, the weighting factor for the $i$-th pair is given by:

$$\text{NECC}(i) = \frac{\exp(C(\mathbf{E}_i, \widetilde{\mathbf{E}}_i))}{\sum_{k=1}^{n} \exp(C(\mathbf{E}_k, \widetilde{\mathbf{E}}_k))}. \tag{12}$$

Treating the original vectors as reference units, NECC is proportional to the extent to which each transformed vector correlates with its corresponding original vector. Finally, employing two components of our vector transformation measurement, we define the transformation weights $\mathbf{W}$:

$$\mathbf{W} = \text{diag}\left(\frac{\text{L}(\widetilde{\mathbf{E}}_1)}{\text{L}(\mathbf{E}_1)}\text{NECC}(1), \frac{\text{L}(\widetilde{\mathbf{E}}_2)}{\text{L}(\mathbf{E}_2)}\text{NECC}(2), \ldots, \frac{\text{L}(\widetilde{\mathbf{E}}_n)}{\text{L}(\mathbf{E}_n)}\text{NECC}(n)\right), \tag{13}$$

and the update map $\mathbf{U}$ for MHSA, incorporating both attention and transformation information:

$$\mathbf{U} = \mathbf{O} + \mathbb{E}_{\text{h}}\left[(\nabla_{\mathbf{A}} p(c))^+ \odot \mathbf{T}\right], \tag{14}$$

with

$$\mathbf{O} = \mathbf{I}, \quad \mathbf{T} = \mathbf{A} \cdot \mathbf{W}. \tag{15}$$

Here, $U_{ij}$ denotes the influence of the $j$-th input vector on the $i$-th output vector of the MHSA layer. This formulation balances both length and direction, reflecting how much of the original information is retained or altered in the transformed vectors, to faithfully evaluate the vector contributions.

We now adapt our established approach to the FFN layer. Recall that FFN also involves a weighted sum of original and transformed vectors (see Equation 7). For 'contextualization' in FFN, there is only a single-head and the weighted combination is performed locally. This simpler form can be easily reflected by discarding the gradient-weighted multi-head integration from Equation 14 and changing the attention map $\mathbf{A}$ in Equation 15 to an identity matrix. We formulate the update map for the FFN layer:

$$\mathbf{U} = \mathbf{O} + \mathbf{T}, \tag{16}$$

with

$$\mathbf{O} = \mathbf{I}, \quad \mathbf{T} = \mathbf{I} \cdot \mathbf{W}. \tag{17}$$

Here, $U_{ij}$ denotes the influence of the $j$-th input vector on the $i$-th output vector of the FFN layer. As illustrated by Figure 3, update maps capture the relative effects and will serve as refinements to the overall contribution map as it aggregates across multiple layers.

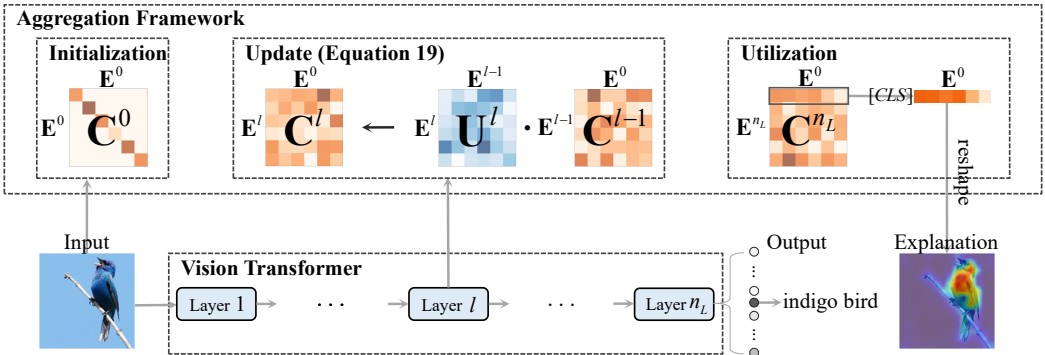

Figure 3: Illustration of our aggregation framework and the explanation pipeline. The overall contribution map is initialized by input vector lengths and is updated using our $\mathbf{U}^l$ to trace vector evolution across layers. In $\mathbf{C}^l$, each $i$-th row represents the influences of input vectors $\mathbf{E}^0$ on the output of the $l$-th layer $\mathbf{E}^l$. For $\mathbf{C}^{n_L}$, the row *w.r.t.* $[CLS]$ vector is extracted and reshaped to produce the final explanation map.

## 3.3 AGGREGATION FRAMEWORK

In Vision Transformers, the influences of vector transformations and contextualizations do not merely reside within isolated layers but accumulate across them. For a comprehensive contribution map that reflects the roles of initial vectors corresponding to input image regions, it is necessary to assess the relationships between input vectors and their evolved counterparts in deeper layers. To this end, we introduce an aggregation framework that cohesively measures the combined influences of contextualization and transformation across the entire model.

### 3.3.1 INITIALIZATION

We denote the overall contribution map by $\mathbf{C} \in \mathbb{R}^{n \times n}$, where $n$ is the number of vectors. The $C_{ij}$ will faithfully accumulate the influence of the $j$-th input vector on the $i$-th output vector. At the initial state, before entering Transformer layers, each input vector simply contains itself without contextualization or transformation. We represent this state using initial vectors' lengths, as depicted in Figure 3. Specifically, we initialize the map as a diagonal matrix $\mathbf{C}^0$:

$$\mathbf{C}^0 = \mathrm{diag}(\mathrm{L}(\mathbf{E}_1^0), \mathrm{L}(\mathbf{E}_2^0), \ldots, \mathrm{L}(\mathbf{E}_n^0)), \tag{18}$$

where $\mathbf{E}^0$ denotes the initial embeddings inputted to the first layer.

### 3.3.2 UPDATE RULES

Based on the DAG representation of information flow [1], layer-wise aggregation is mathematically equivalent to matrix multiplication of intermediate maps. This approach allows for tracing vector contributions throughout the model. After initialization, we iteratively update the map $\mathbf{C}^{l-1}$ using $\mathbf{U}^l$, which incorporates the effects of transformed vectors:

$$\mathbf{C}^l \leftarrow \mathbf{U}^l \cdot \mathbf{C}^{l-1} \quad \text{for} \quad l = 1, 2, \ldots, n_L. \tag{19}$$

In this formula, $\mathbf{C}^{l-1}$ represents the map that has traced the vector contributions up to the $(l-1)$-th layer but has yet to include the effects from the $l$-th layer, and $\mathbf{U}^l$ indicates the update map from the $l$-th layer. Using this recursive formula, the map aggregates information about both vector transformation and contextualization over the entire model. After applying the updates for all the layers, the final contribution map can be expressed as:

$$\mathbf{C}^{n_L} = \mathbf{U}^{n_L} \cdot \mathbf{U}^{n_L-1} \cdot \cdots \cdot \mathbf{U}^1 \cdot \mathbf{C}^0, \tag{20}$$

This framework provides a cumulative understanding of how initial input vectors contribute to the final prediction, thereby faithfully interpreting the decision process of Vision Transformers.

## 3.4 UTILIZING THE OVERALL CONTRIBUTION MAP FOR INTERPRETATION

Vision Transformers often use designated vectors to perform specific tasks. For example, ViT [15] performs image classification based on a $[CLS]$ vector. This special vector becomes significant

| Grad-CAM | Trans. Attr. | Raw Att. | Rollout | Trans. MM | **Ours** |
|---|---|---|---|---|---|

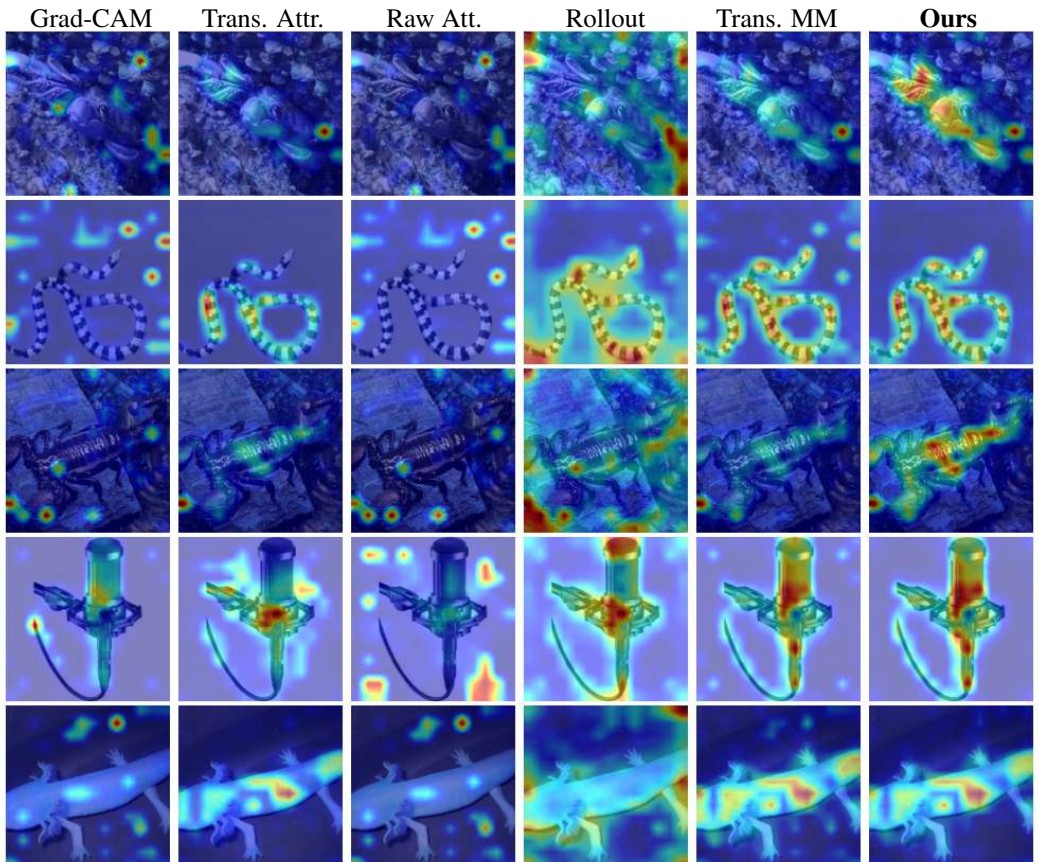

Figure 4: Visualizations of explanation results. Our method provides more object-centric heatmaps.

in the final prediction as it integrates information from all the inputs. To interpret the model's prediction, one can analyze the contribution of each initial input vector to the final $[CLS]$. In our proposed VTranM, this is embodied in the overall contribution map $\mathbf{C}^{n_L}$. Specifically, the row in $\mathbf{C}^{n_L}$ that corresponds to the $[CLS]$ vector represents the influences of original vectors. As illustrated in Figure 3, we extract this row and reshape it to the image's spatial dimensions, forming a contribution heatmap. This heatmap highlights regions that are highly influential in determining the prediction, serving as a post-hoc explanation of the model's decision.

## 4 EXPERIMENTS

In this section, we first introduce experimental setups. Then we demonstrate the superiority of our VTranM qualitatively and quantitatively. Lastly, we take an ablation study on the proposed method.

### 4.1 EXPERIMENTAL SETTINGS

#### 4.1.1 BASELINE METHODS

We consider baseline methods that are widely used and applicable from three types. **(i)** Gradient-based: Grad-CAM [35], **(ii)** Attribution-based: LRP [7], Conservative LRP [3], and Transformer Attribution [11]), and **(iii)** Attention-based: Raw Attention [24], Rollout [1], ATTCAT [33], and Transformer MM [10]). Details of experimental setups are provided in Appendix B.

#### 4.1.2 EVALUATED PROPERTIES

**Localization Ability.** This property measures how well an explanation can localize the foreground object recognized by the model. Intuitively, a reliable explanation should be object-centric, *i.e.*, accurately highlighting the object that the model uses to make the decision. Following previous works

Table 1: Results of Impact on Accuracy. We report results on CIFAR-10 and CIFAR-100 [26].

| | Neg. (Pred.) ↑ | Pos. (Pred.) ↓ | Neg. (GT) ↑ | Pos. (GT) ↓ |
|---|---|---|---|---|
| Grad-CAM [35] | 61.24 / 47.18 | 47.32 / 41.91 | 61.26 / 47.28 | 47.34 / 41.78 |
| LRP [7] | 72.62 / 46.76 | 72.14 / 46.31 | 72.54 / 46.80 | 72.13 / 46.29 |
| Cons. LRP [3] | 44.92 / 27.11 | 49.15 / 27.99 | 45.76 / 26.84 | 47.87 / 27.42 |
| Trans. Attr. [11] | 67.65 / 44.65 | 43.64 / 24.99 | 65.82 / 44.19 | 43.17 / 23.90 |
| Raw Att. [24] | 58.46 / 37.23 | 47.17 / 28.62 | 58.46 / 37.23 | 47.17 / 28.62 |
| Rollout [1] | 64.88 / 41.34 | 42.89 / 24.96 | 64.88 / 41.34 | 42.89 / 24.96 |
| ATTCAT [33] | 41.78 / 23.52 | 42.66 / 22.15 | 41.91 / 23.90 | 42.52 / 21.73 |
| Trans. MM [10] | 71.67 / 52.14 | 37.77 / 17.52 | 71.81 / 52.65 | 37.70 / 17.32 |
| **Ours** | **74.90** / **54.78** | **37.14** / **16.07** | **75.05** / **55.30** | **36.48** / **15.89** |

Table 2: Results of Localization Ability and Impact on Accuracy on Imagenet [34].

| | Localization Ability | | | Impact on Accuracy | | | |
|---|---|---|---|---|---|---|---|
| | Pix. Acc. ↑ | mAP ↑ | mIoU ↑ | Neg. (Pred.) ↑ | Pos. (Pred.) ↓ | Neg. (GT) ↑ | Pos. (GT) ↓ |
| Grad-CAM [35] | 67.37 | 79.20 | 46.13 | 43.84 | 26.11 | 44.06 | 25.87 |
| LRP [7] | 50.96 | 55.87 | 32.82 | 42.55 | 41.29 | 42.56 | 41.29 |
| Cons. LRP [3] | 64.06 | 68.01 | 36.23 | 33.24 | 34.35 | 34.33 | 34.03 |
| Trans. Attr. [11] | 79.70 | 86.03 | 61.95 | 57.48 | 19.00 | 58.26 | 18.38 |
| Raw Att. [24] | 67.87 | 80.24 | 46.37 | 48.70 | 25.44 | 48.70 | 25.44 |
| Rollout [1] | 59.01 | 73.76 | 39.43 | 43.23 | 32.34 | 43.23 | 32.34 |
| ATTCAT [33] | 43.88 | 48.38 | 27.90 | 32.34 | 31.71 | 33.59 | 30.68 |
| Trans. MM [10] | 79.05 | 85.71 | 61.56 | 57.53 | 15.88 | 58.85 | 15.20 |
| **Ours** | **80.53** | **86.33** | **63.45** | **58.12** | **15.75** | **59.36** | **15.09** |

[35, 6, 11, 10], we perform segmentation on the ImageNet-Segmentation dataset [19], using DeiT [44], a prevalent Vision Transformer model. We regard the explanation heatmaps as preliminary semantic signals. Then we use the average value of each heatmap as a threshold to produce binary segmentation maps. Based on the ground-truth maps, we evaluate the performance on three metrics: Pixel-wise Accuracy, mean Intersection over Union (mIoU), and mean Average Precision (mAP).

**Impact on Accuracy.** This aspect focuses on how well an explanation captures the correlations between pixels and the model's accuracy. We assess this property by perturbation tests on CIFAR-10, CIFAR-100 [26], and ImageNet [34]. The pre-trained DeiT model is used to perform both positive and negative perturbation tests *w.r.t.* the predicted and the ground-truth class. Specifically, these tests include two stages. First, we produce explanation maps for the class we want to visualize. Second, we gradually remove pixels and evaluate the resulting mean top-1 accuracy. In positive tests, pixels are removed from the most important to the least, while in negative tests, the removal order is reversed. A faithful explanation should assign higher scores to pixels that contribute more. Thus, a severe drop in the model's accuracy is expected in positive tests, while we expect the accuracy to be maintained in negative tests. For quantitative evaluation, we compute the Area Under the Curve (AUC) [4] *w.r.t.* the accuracy curve of different perturbation levels.

**Impact on Probability.** This property further measures how well the explanations illustrate important pixels for the model's predicted probabilities. Similarly, it is evaluated by perturbation tests on ImageNet. Instead of AUC, we report the Area Over the Perturbation Curve (AOPC) and the Log-odds score (LOdds) [37, 32, 12]. These metrics quantify the average change of output probabilities *w.r.t.* the predicted label. For comprehensive evaluations, we report results on ViT-Large [15] together with DeiT [44] and ViT-Base [15].

## 4.2 EXPERIMENTAL RESULTS

**Qualitative Evaluation.** As shown by Figure 4, our proposed VTranM can generate more precise and exhaustive heatmaps. Appendix A provides more visualizations.

**Quantitative Evaluation. (i) Localization Ability.** Table 2 (left) reports the segmentation results on the ImageNet-Segmentation dataset. Our proposed VTranM significantly outperforms all the baselines on pixel accuracy, mIoU, and mAP, demonstrating its stronger localization ability. **(ii) Impact on Accuracy.** Table 1 and Table 2 show the results of the perturbation tests for both classes. For positive tests, a lower AUC indicates superior performance, while for negative tests, a higher AUC is desirable. The results underscore the superiority of our method. **(iii) Impact on Probability.** Table 3 reports the results of perturbation tests assessing the output probability. Our VTranM achieves the best performance on three Vision Transformers variants.

Table 3: Results of Impact on Probability on three Vision Transformer models.

| | AOPC ↑ | | | LOdds ↓ | | |
|---|---|---|---|---|---|---|
| | DeiT [44] | ViT-B [15] | ViT-L [15] | DeiT [44] | ViT-B [15] | ViT-L [15] |
| Grad-CAM [35] | 0.253 | 0.557 | 0.457 | -3.704 | -4.020 | -3.023 |
| LRP [7] | 0.180 | 0.474 | 0.508 | -2.140 | -3.259 | -3.634 |
| Cons. LRP [3] | 0.218 | 0.613 | 0.614 | -3.011 | -4.272 | -4.443 |
| Trans. Attr. [11] | 0.282 | 0.721 | 0.716 | -4.163 | -5.622 | -5.541 |
| Raw Att. [24] | 0.254 | 0.652 | 0.655 | -3.657 | -4.911 | -4.961 |
| Rollout [1] | 0.229 | 0.688 | 0.689 | -3.223 | -5.259 | -5.183 |
| ATTCAT [33] | 0.235 | 0.641 | 0.645 | -3.262 | -4.461 | -4.622 |
| Trans. MM [10] | 0.296 | 0.719 | 0.719 | -4.504 | -5.567 | -5.623 |
| **Ours** | **0.297** | **0.725** | **0.725** | **-4.514** | **-5.676** | **-5.725** |

**Ablation Study.** We conduct ablation studies on the effect of our proposed vector transformation measurement (L and NECC) and aggregation framework (AF). We perform experiments on ImageNet, based on ViT. Four variants are considered: **(i)** Baseline, which merely applies Equation 8 to the input without aggregation, **(ii)**

Table 4: Ablation study on the proposed method.

| | Segmentation | | | Perturbation Test | |
|---|---|---|---|---|---|
| | Pix. Acc. ↑ | mAP ↑ | mIoU ↑ | Neg. ↑ | Pos. ↓ |
| Baseline | 52.94 | 67.78 | 33.42 | 44.92 | 31.95 |
| + AF | 76.30 | 85.28 | 58.34 | 55.25 | 16.98 |
| + AF + L | 77.50 | 85.39 | 59.59 | 55.03 | 16.28 |
| + AF + L + NECC | **77.89** | **85.69** | **60.29** | **55.42** | **16.18** |

Baseline + AF, which initializes the contribution map as an identity matrix and aggregates using $\mathbf{U}^l$, where relative lengths and NECC are discarded, **(iii)** Baseline + AF + L, which additionally use relative length measurement in $\mathbf{U}^l$ while still excluding the NECC, and **(iv)** Baseline + AF + L + NECC, which is our VTranM. As shown in Table 4, each proposed component improves the performance, validating the effectiveness.

## 5 RELATED WORK

**Traditional post-hoc explanations.** General traditional post-hoc explanation methods mainly fall into two groups: gradient-based and attribution-based. Examples of gradient-based methods are Gradient*Input [38], SmoothGrad [39], Deconvolutional Network [48], Full Grad [41], Integrated Gradients [43], and Grad-CAM [35]. These methods produce saliency maps using the gradient. On the other hand, attribution-based methods propagate classification scores backward to the input [5, 29, 37, 23, 18, 20], based on Deep Taylor Decomposition [31]. There are other approaches beyond these two types, such as saliency-based [49, 30, 13], Shapley additive explanation (SHAP) [29], and perturbation-based methods [17]. Although initially designed for MLPs and CNNs, some traditional methods have been adapted for Transformers in recent works [11, 3]. However, without essential attention information, these methods still yield suboptimal performance on Vision Transformers.

**Transformer-specific attention-based explanations.** A growing line of work in interpretability develops new paradigms specifically for Transformers. Attention maps are widely used in this direction, as they are intrinsic distributions of scaling weights over vectors. Representative methods include Raw Attention [47], a method simply employing the inherent attention information, Rollout [1], which linearly accumulates attention maps, Transformer-MM [10], a general framework for cross-attention and encoder-decoder Transformers, and ATTCAT [33], a method that formulates Attentive Class Activation Tokens to estimate the relative importance among input vectors. However, these approaches overlook the relative effects of transformations and fail to faithfully aggregate vector contributions across all modules, hindering reliable interpretations of Vision Transformers.

## 6 CONCLUSIONS

In this paper, we explored the challenge of explaining Vision Transformers. Upon reinterpretation, we expressed Transformer layers using a generic form in terms of weighted linear combinations of original and transformed vectors. This new perspective revealed a principal issue in existing methods: they solely rely on attention weights and ignore significant information about vector transformations, thus failing to explain the models' inner workings. To tackle this problem, we proposed VTranM, an explanation method comprising a vector transformation measurement and an aggregation framework. Faithfully evaluating the contribution of input vectors throughout the entire model, our method offers more object-centric interpretations.

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

# A    MORE VISUALIZATION RESULTS

Figure 5 illustrates the explanations produced by various methods. Compared to the baselines, our explanations are more accurate in terms of both background regions and foreground objects.

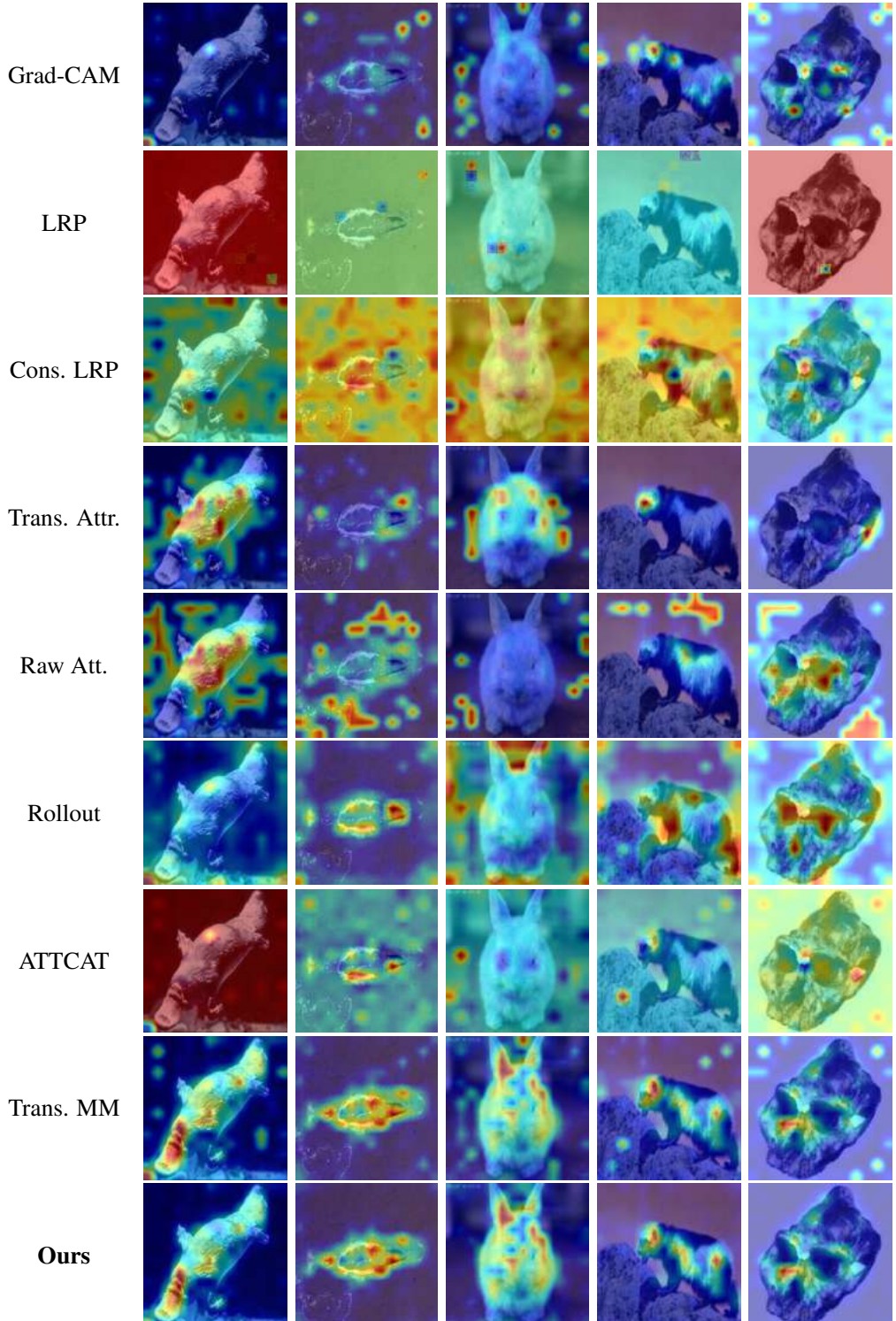

Figure 5: Visualizations of explanation results.

## B DETAIL OF EXPERIMENTAL SETUP

### B.1 DATASETS

**CIFAR-10 and CIFAR-100.** CIFAR-10 and CIFAR-100 [26] are two widely used image classification datasets, each containing 60,000 $32 \times 32$ color images. CIFAR-10 has 10 classes, while CIFAR-100 has a more challenging setting with 100 classes. Both datasets are split into 50,000 training and 10,000 testing images. In this paper, we evaluate explanation methods on validation sets.

**ImageNet.** ImageNet [34] is a large-scale benchmark in the field of image classification. In this work, we evaluate explanation methods on the validation set, which comprises 50,000 high-resolution images across 1,000 distinct classes. Each class contains roughly the same number of images, ensuring a balanced benchmark.

**ImageNet-Segmentation.** ImageNet-Segmentation [19] is an annotated subset of ImageNet, containing 4,276 images from 445 categories.

### B.2 IMPLEMENTATION OF BASELINE METHODS

#### B.2.1 GRADIENT-BASED METHODS

**Grad-CAM.** The Grad-CAM method [35] considers the last attention map and utilizes the row corresponding to the $[CLS]$ vector, which is then reshaped to the 2D image space. Different from Raw Attention, Grad-CAM performs multi-head integration based on gradient information. We implement this method on Vision Transformers following previous works [11, 10].

#### B.2.2 ATTRIBUTION-BASED METHODS

**LRP.** LRP [7] starts from the model's output and propagates relevance scores backward up to the input image. This propagation adheres to a set of rules defined by the Deep Taylor Decomposition theory [31].

**Conservative LRP.** Conservative LRP [3] introduces specialized Layer-wise Relevance Propagation rules for attention heads and layer norms in Transformer models. This is designed to implement conservation, a common property of attribution techniques.

**Transformer Attribution.** Transformer Attribution [11] is an attribution method specifically designed for Transformer models. It first computes relevance scores via modified LRP, and then integrates these scores with attention maps to produce an explanation.

#### B.2.3 ATTENTION-BASED METHODS

**Raw Attention.** This method [24] extracts the multi-head attention map from the last layer of the model and reshapes the row corresponding to the $[CLS]$ vector into the 2D image space. The explanation result is further obtained by averaging across different heads.

**Rollout.** Rollout [1] interprets the information flow within Transformers from the perspective of Directed Acyclic Graphs (DAGs). It traces and accumulates the attention weights across various layers using a linear combination strategy.

**ATTCAT.** ATTCAT [33] is a Transformer explanation technique using attentive class activation tokens. It employs a combination of encoded features, their associated gradients, and their attention weights to produce confident explanations.

**Transformer-MM.** Transformer-MM [10] is a general interpretation framework applicable to diverse Transformer architectures. It aggregates attention maps with corresponding gradients to generate class-specific explanations.

### B.3 EVALUATION METRICS

**Area Under the Curve (AUC) ↓.** This metric calculates the Area Under the Curve (AUC) corresponding to the model's performance as different proportions of input pixels are perturbed [4]. To elaborate, we first generate new data by gradually removing pixels in increments of 5% (from 0% to 100%) based on their explanation weights. The model's accuracy is then assessed on these perturbed data, resulting in a sequence of accuracy measurements. The AUC is subsequently computed using this sequence.

**Area Over the Perturbation Curve (AOPC) ↑.** AOPC [32, 12] measures the changes in output probabilities *w.r.t.* the predicted label after perturbations:

$$\text{AOPC} = \frac{1}{|\mathbb{K}|} \sum_{k \in \mathbb{K}} (\hat{p}(y|\mathbf{x}) - \hat{p}(y|\mathbf{x_k})), \tag{21}$$

where $\mathbb{K} = \{0, 5, ..., 95, 100\}$ is a set of perturbation levels, $\hat{p}(y|\mathbf{x})$ estimates the probability for the predicted class given a sample $\mathbf{x}$, and $\mathbf{x_k}$ is the perturbed version of $\mathbf{x}$, from which the top $k\%$ elements ranked by explanation weights are eliminated.

**Log-odds score (LOdds) ↓.** LOdds [37, 33] averages the difference between the negative logarithmic probabilities on the predicted label before and after masking $k\%$ top-scored pixels over the perturbation set $\mathbb{K}$:

$$\text{LOdds} = -\frac{1}{|\mathbb{K}|} \sum_{k \in \mathbb{K}} \log \frac{\hat{p}(y|\mathbf{x})}{\hat{p}(y|\mathbf{x_k})}. \tag{22}$$

The notations are the same as in Equation 21.

