# OpenReview forum: "VTranM: Vision Transformer Explainability with Vector Transformations Measurement"
_ICLR.cc/2024/Conference — ICLR 2024 Conference Withdrawn Submission_

### Official Review · Reviewer_NcNK · 2023-11-01

**Soundness:** 3 good
**Presentation:** 3 good
**Contribution:** 3 good
**Rating:** 6
**Confidence:** 4

**Summary:**

The paper proposed an explanation method leveraging vector transformation measurement.
It evaluates transformation effects by considering changes in vector length and directional correlation.
It further incorporates attention and vector transformation information across layers to capture the comprehensive vector contributions over the entire model.
Experiments demonstrate good explanation performance.

**Strengths:**

+ The proposed method is clear and novel. The implementation is sound.
+ It provides an aggregation framework to trace vector evolution across layers.
+ It demonstrates better visual results on  object-centric heatmaps.
+ Numeric studies also prove its advantages over traditional methods.

**Weaknesses:**

- The method only demonstrate the aggregation method on plain ViT. I am concerned that it will not work on other vision transformers with window / shifting attentions

**Questions:**

Is proposed method compatible with hybird vision transformers with convolution layers? Consider to re-run table 3 on more networks, such as Swin and PVT

---

### Official Review · Reviewer_9oRD · 2023-11-04

**Soundness:** 3 good
**Presentation:** 3 good
**Contribution:** 2 fair
**Rating:** 5
**Confidence:** 5

**Summary:**

This work introduces an explanation method for the vision transformer.  The main idea is taking the changes in vector length and direction into consideration. Then the author builds an aggregation framework for understanding the vision transformer. The empirical results indicate that the proposed method is helpful for improving the performance of the intermediate prediction on classification and localization.

**Strengths:**

1. The paper presents a straightforward and intuitive approach by formulating the MLP and FFN as the vectors transformation, making it easy to understand. The analytical process is clearly outlined.
2. The experimental section is thorough, with a wide range of tasks being evaluated.
3. The proposed aggregation framework appears to be both simple and efficient.

**Weaknesses:**

1. The main concern is the extent to which this method and framework will have an impact.  It seems the proposed method just offers a new way to visualize the highlighted regions in a ViT.
2. The analysis is currently limited to ViTs trained using supervised image classification.
3. The explanation section is lacking, leaving it unclear whether any new insights about ViTs have been gained through this framework.
4. Many neural network architectures can be understood as vector transformations, including LSTM, RNN and CNN). Therefore, the novelty and originality of this work should be more thoroughly discussed.

**Questions:**

1. While the proposed method offers a new visualization tool for ViTs, it remains unclear how it can help us better understand ViTs or if it provides any novel insights into their workings.
2. How to use this method to analyze ViTs trained with self-supervised learning, like the DINO v1/v2, MAE, etc. Please prodive more insights about these models with the proposed method.
3. What's the difference between the ViTs and CNNs when understanding the network from the vector transformation perspective.

---

### Official Review · Reviewer_WGBd · 2023-11-06

**Soundness:** 3 good
**Presentation:** 3 good
**Contribution:** 2 fair
**Rating:** 3
**Confidence:** 3

**Summary:**

This paper presents VTranM, an explanation method for Vision Transformers that addresses the limitations of current explanation methods. While Vision Transformers draw representations from image patches as transformed vectors and integrate them using attention weights, current explanation methods only focus on attention weights without considering essential information from the corresponding transformed vectors. To accommodate the contributions of transformed vectors, the authors propose VTranM, which leverages a vector transformation measurement that faithfully evaluates transformation effects by considering changes in vector length and directional correlation. Furthermore, they use an aggregation framework to incorporate attention and vector transformation information across layers, thus capturing the comprehensive vector contributions over the entire model. The authors demonstrate the superiority of VTranM compared to state-of-the-art explanation methods in terms of localization ability, segmentation, and perturbation tests. Their experiments show that VTranM produces more accurate explanations in terms of both background regions and foreground objects. Overall, this work provides contributions to the field of vision transformers by introducing an explanation method that can improve the interpretability and transparency of Vision Transformers.

**Strengths:**

- This work introduces an explanation method, VTranM, that addresses the limitations of current explanation methods for Vision Transformers. The proposed vector transformation measurement and aggregation framework improve the performance of visualization.

- The authors demonstrate the superiority of VTranM compared to state-of-the-art explanation methods in terms of localization ability, segmentation, and perturbation tests. Their experiments show that VTranM produces more accurate explanations in terms of both background regions and foreground objects.

-  The authors conduct a comprehensive evaluation of VTranM, including qualitative and quantitative analyses, ablation studies, and comparisons with baseline methods. This evaluation provides a thorough understanding of the strengths and weaknesses of VTranM.

**Weaknesses:**

- The improvement is very limited. From Tables 1 and 2, the improvement is marginal compared to Trans. MM.

- Lack of analysis of the proposed method. What if the proposed module is applied to only one of these transformer blocks, with different positions? How will it affect the results?

- Lack of interpretability. While the proposed method provides more accurate explanations than baseline methods, it is still unclear how well the explanations reflect the underlying decision-making process of the model.

- Missing discussions with [a, b]

Missing References:
[a] MemNet: A Persistent Memory Network for Image Restoration
[b] IA-RED2: Interpretability-Aware Redundancy Reduction for Vision Transformers

**Questions:**

- How is the proposed method compared to DINO-v1/-v2 [c, d]? It seems that the visual results are much worse than DINO-v2. The self-supervised training method could be also seen as another explanability framework if its linear head is finetuned on the target dataset.

References:
[c] Emerging Properties in Self-Supervised Vision Transformers
[d] DINOv2: Learning Robust Visual Features without Supervision